# The chemistry of branched condensed phosphates

Tobias Dürr-Mayer[1], Danye Qiu [1], Verena B. Eisenbeis [1], Nicole Steck [1], Markus Häner [1],
Alexandre Hofer [2], Andreas Mayer [3], Jay S. Siegel [4,5], Kim K. Baldridge[4] & Henning J. Jessen [1,5,6✉]

Condensed phosphates may exist as linear, cyclic or branched structures. Due to their important role in nature, linear polyphosphates have been well studied. In contrast, branched phosphates (ultraphosphates) remain largely uncharacterised, because they were already described in 1950 as exceedingly unstable in the presence of water, epitomized in the antibranching-rule. This rule lacks experimental backup, since, to the best of our knowledge, no rational synthesis of defined ultraphosphates is known. Consequently, detailed studies of their chemical properties, reactivity and potential biological relevance remain elusive. Here, we introduce a general synthesis of monodisperse ultraphosphates. Hydrolysis half-lives up to days call the antibranching-rule into question. We provide evidence for the interaction of an enzyme with ultraphosphates and discover a rearrangement linearizing the branched structure. Moreover, ultraphosphate can phosphorylate nucleophiles such as amino acids and nucleosides with implications for prebiotic chemistry. Our results provide an entry point into the uncharted territory of branched condensed phosphates.

[1] Institute of Organic Chemistry, University of Freiburg, Freiburg, Germany. [2] Department of Chemistry, University of Cambridge, Cambridge, United Kingdom.
[3] Département de Biochimie, Université de Lausanne, Epalinges, Switzerland. [4] Health Science Platform, Tianjin University, Tianjin, P R China. [5] Freiburg
Research Institute for Advanced Studies, University of Freiburg, Freiburg, Germany. [6] Cluster of Excellence livMatS @ FIT – Freiburg Center for Interactive
Materials and Bioinspired Technologies, University of Freiburg, Freiburg, Germany. ✉email: henning.jessen@oc.uni-freiburg.de

Polyphosphates (polyP) are polymers of orthophosphate linked by phosphoanhydride bonds. They are ubiquitous in living organisms with numerous biological functions[1]. PolyP may exist in three principally different structures (see Fig. 1a): linear, cyclic (metaphosphates) or branched (ultraphosphates)[2]. Cellular polyP are now defined as exclusively linear polymers[1, 3–5]. This paradigm evolved, despite early reports on the presence of metaphosphates in cellular extracts[6–8] and was recently called into question by $^{31}$P solid-state NMR data from whole *Xanthobacter autotrophicus*[9]. There have only been scattered comments on ultraphosphates in biology[3, 10]. These, in turn, dismiss the occurrence of ultraphosphates by referring to the antibranching-rule, which was coined in 1950 and has persisted since then[11–15]. Some work has been done on vitreous and crystalline ultraphosphates[16–22] owing to applications as laser materials (see Fig. 1b)[11, 23]. Yet, studies on monodisperse ultraphosphates are limited. In the 1970s, Glonek reported cyclic ultraphosphate structures from condensations of ortho- or metaphosphates (see Fig. 1c)[24–28]. The ultraphosphates could neither be obtained in pure form nor isolated and in part lack unambiguous analytical proof. The addition of water resulted in the instantaneous hydrolysis of the branches[24].

In recent years, branched oligophosphates caught attention again: cyclic ultraphosphates like **11** with modifications on the terminal phosphate can now be obtained and applied in the syntheses of linear nucleoside, dinucleoside and inorganic polyP[29–33]. Cummins et al. synthesised **7** as its [PPN] (bis(-triphenylphosphine)iminium) salt, which allowed the isolation of the product[34]. Application of **7** was demonstrated in the tetra-phosphorylation of nucleotides and enabled the synthesis of another ultraphosphate species containing four phosphates in the cyclic subunit[35]. Although cyclic ultraphosphates have thus recently become accessible, there is no synthesis of ultraphosphates available that are devoid of cyclic substructures and therefore are true constitutional isomers of linear polyP[36]. This fundamental type of condensed phosphates remained unstudied.

Here, we report the synthesis of non-cyclic ultraphosphates using phosphoramidite chemistry. This approach provides access to both symmetrically modified ultraphosphates containing three equal modifications on each terminal phosphate, including the smallest possible unmodified ultraphosphate uP$_4$ (**2**) as well as unsymmetrical analogues containing two different residues. The synthetic approach enables the generation of thio- and seleno-ultraphosphates. To interrogate the antibranching-rule, we report hydrolysis studies including enzymatic degradation. Furthermore, we show the phosphorylation of nucleosides with inorganic ultraphosphate with implications for prebiotic chemistry (see Fig. 1d). We also study the reactivity of modified ultraphosphates in organic solvent by using a combination of $^{31}$P-NMR and capillary electrophoresis mass spectrometry (CE-MS) and discover an ultraphosphate rearrangement, which we name the phosphate walk.

## Results

**Symmetrical ultraphosphates**. Phosphordiamidites are used in linear polyP syntheses by twofold activation and reaction with (modified) phosphates[37]. A phosphortriamidite should therefore enable threefold activation and reaction with three phosphates to produce a mixed P(III)-P(V)-anhydride intermediate **18** (hereafter, called ultraphosphite, Fig. 2a), which can further be oxidised resulting in an ultraphosphate (**12**).

Initially, the reaction of three equivalents of tetrabutylammonium (TBA) phenyl phosphate with tris(diethylamino)phosphine (**16**) and ethylthiotetrazole (ETT) was studied: $^{31}$P{$^1$H}-NMR of the mixture showed consumption of **16** within a few minutes,

giving rise to the ultraphosphite intermediate. The absence of peak splitting due to homonuclear P-P coupling for this mixed P(III)-P(V)-anhydride is in accordance with earlier observations for P-amidite couplings[37]. Oxidation with *m*CPBA gave phenyl-modified ultraphosphate **20**, which was isolated by precipitation with Et$_2$O (71% yield, 78% purity, measured by $^{31}$P{$^1$H}-NMR). As decomposition products, diphenyl triphosphate (10%) and phenyl phosphate (10%) were detected. The central ultraphosphate signal, now showing the expected multiplicity (quartet) with a chemical shift of ca. $\delta = -35$ ppm, was detected in water. This finding is in sharp contrast to the antibranching-rule, claiming the instantaneous hydrolysis of ultraphosphates[3, 10, 24]. Purification was possible in aqueous buffer (NH$_4$HCO$_3$) using strong anion exchange chromatography (SAX) and $^{31}$P{$^1$H}-NMR analysis of fractions showed pure **20** in solution. Lyophilization resulted in significant decomposition. We also studied NaClO$_4$ or LiCl solutions as eluents in SAX and attempted to isolate the product by precipitation from acetone[32]. Although we were unable to isolate trisphenyl uP$_4$ **20** using this procedure, trisadenosine ultraphosphate (**21**) precipitated and afforded the product in 55% yield. $^{31}$P{$^1$H}-NMR of the precipitate showed pure ultraphosphate, but after drying, decomposition (>15%) was detected.

Next, we evaluated the scope of ultraphosphate synthesis (see Fig. 2b) by changing modifications on the terminating phosphates. Since not all products precipitated readily, reaction yields as derived for **21** cannot be given and Fig. 2b lists the purities of crude products according to $^{31}$P{$^1$H}-NMR. Alkyne modified ultraphosphates (**24** and **25**) were readily accessible. Efforts towards the synthesis of amino-acid- and carbohydrate-modified ultraphosphates using *O*-phospho-L-tyrosine or α-D-glucose-1-phosphate resulted in side-reactions of the phosphoramidite with the amine or the primary alcohol, respectively. Application of Fmoc-*O*-phospho-L-tyrosine followed by deprotection of the amine gave access to amino-acid modified ultraphosphate **27**. The use of D-glucose-6-phosphate enabled access to a carbohydrate-containing ultraphosphate **28**. Thiamine-derived ultraphosphate **29** could be obtained from the tetrakis[3,5-bis(trifluoromethyl)phenyl]borate salt of thiamine phosphate. Modification of the oxidation (S$_8$ or KSeCN) facilitated entry into thio- (in green) and seleno- (**23**) ultraphosphates.

**Unmodified and unsymmetrical ultraphosphates**. We envisioned the synthesis of inorganic ultraphosphate uP$_4$ (**2**) as the defining minimal unit of this substance class. Reactions of phosphoric acid with **16** were unsuccessful thus requiring protected precursors for **2**. Different cleavage strategies for the protected phosphates were considered, including hydrogenolytic (**30**), enzymatic (**31**, discussed later), phototriggered (**32–36**) and basic (**37**) deprotection (see Fig. 2b). We found that (9*H*-fluorenyl-9-yl)methyl (Fm) dihydrogen phosphate was readily accessible[38] and that the corresponding ultraphosphate **37** could be synthesised under ambient conditions. **37** could also be stored indefinitely in solution at −20 °C after purification, enabling screening of several bases for deprotection. Only DBU enabled deprotection to **37**, but precipitation of uP$_4$ **2** resulted in decomposition.

Unsymmetrically modified ultraphosphates were accessible by orthogonal activation strategies using chlorophosphoramidites (**39** or **41**, see Fig. 3a). Although over-reaction in the first and unselective phosphate exchange in the second step was observed, a series of unsymmetrical ultraphosphates were obtained after purification. Fm-modified structures further allowed deprotection by the addition of DBU to yield twofold or singly modified ultraphosphates (**50–54**, see Fig. 3c). The number of

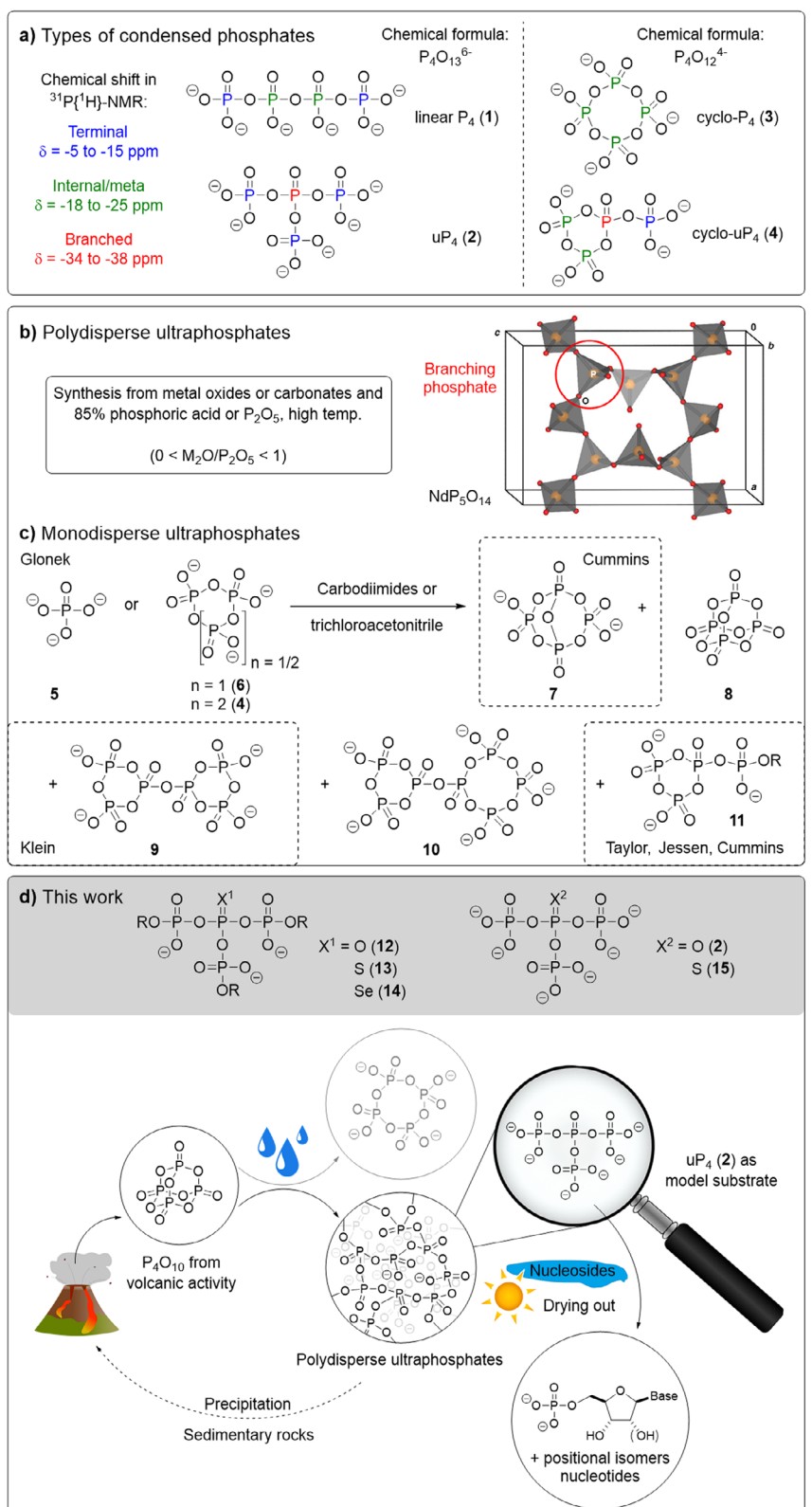

**Fig. 1 Previous work on cyclic ultraphosphates and non-cyclic ultraphosphates in this work with implications for prebiotic chemistry. a** Types of condensed phosphates and their chemical shift in $^{31}P\{^{1}H\}$-NMR. **b** General synthesis protocol for polydisperse ultraphosphates and crystal structure of $NdP_5O_{14}$[61]. **c** Monodisperse ultraphosphates detected by Glonek in the condensation reactions of orthophosphoric acid or metaphosphates using carbodiimides or trichloroacetonitrile[24–28]. The structures in dashed boxes were further described in publications by Klein[62], Taylor[29], Jessen[31, 32] and Cummins[33]. **d** General structures of ultraphosphates in this work and suggestion for prebiotically plausible phosphorylation reactions using ultraphosphates: $uP_4$ (**2**) was used as a model substrate for polydisperse ultraphosphates arising from reactions of $P_4O_{10}$ in the presence of water[59]. The prebiotic phosphate cycle including phosphorus pentoxide from volcanic activity has already been proposed and meets the challenge of making phosphate available from low-solubility minerals[56–58, 63].

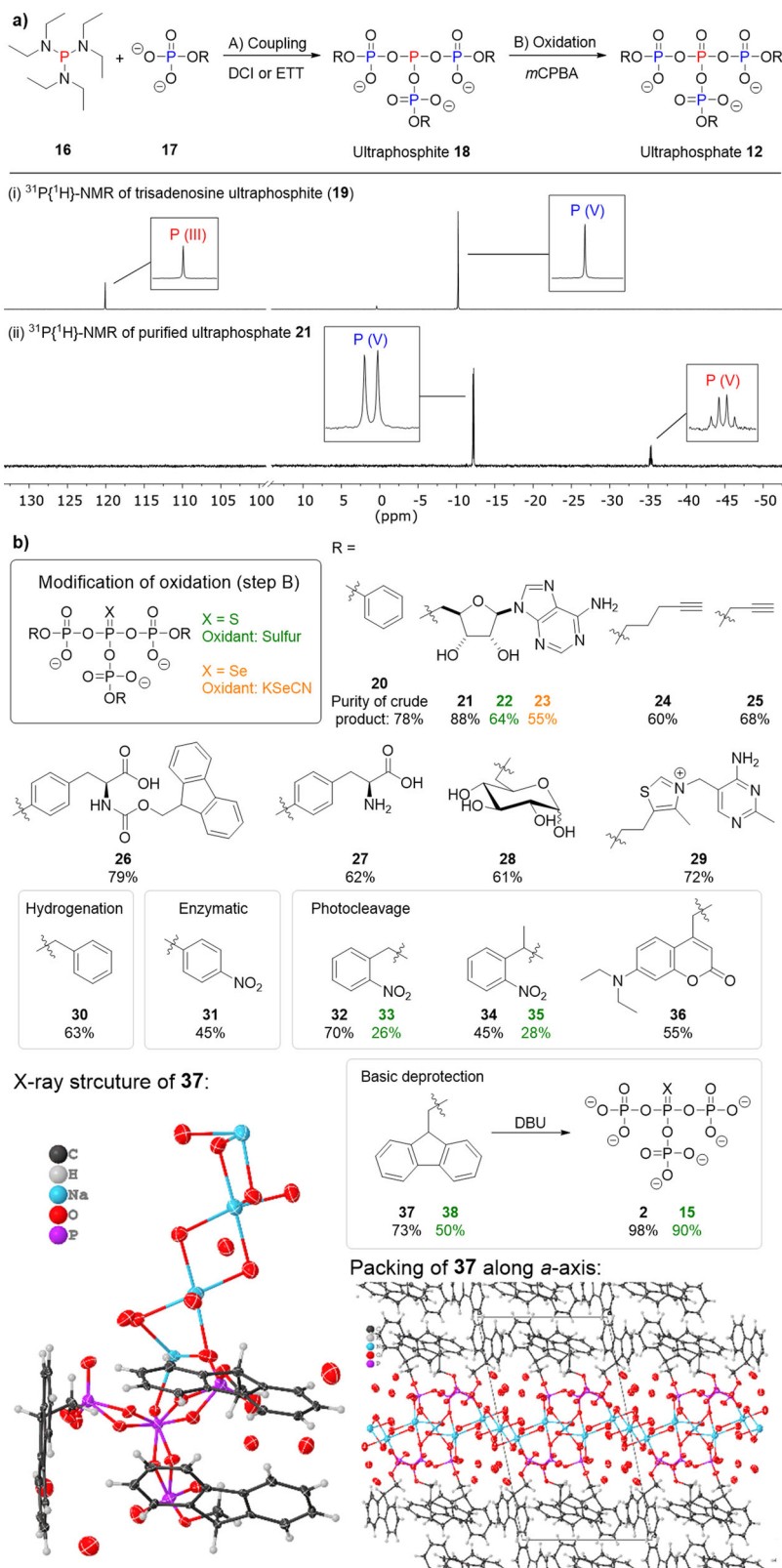

**Fig. 2 Symmetrical ultraphosphates. a** Synthesis of ultraphosphates by threefold coupling of phosphates with tris(diethylamino)phosphine **16** and subsequent oxidation. $^{31}$P{$^1$H}-NMR spectra of the intermediary ultraphosphite **19** and the ultraphosphate **21** after oxidation and purification for R = adenosine. **b** Synthetic scope of the synthesis of symmetrically modified ultraphosphates and modification of the oxidation step to yield thio- and seleno analogues. The purity of the crude products is given in percent according to the $^{31}$P{$^1$H}-NMR spectra. Yields were not determined due to uncertain quantities of counterions but the respective crude masses are reported in the supporting information along with the spectra.

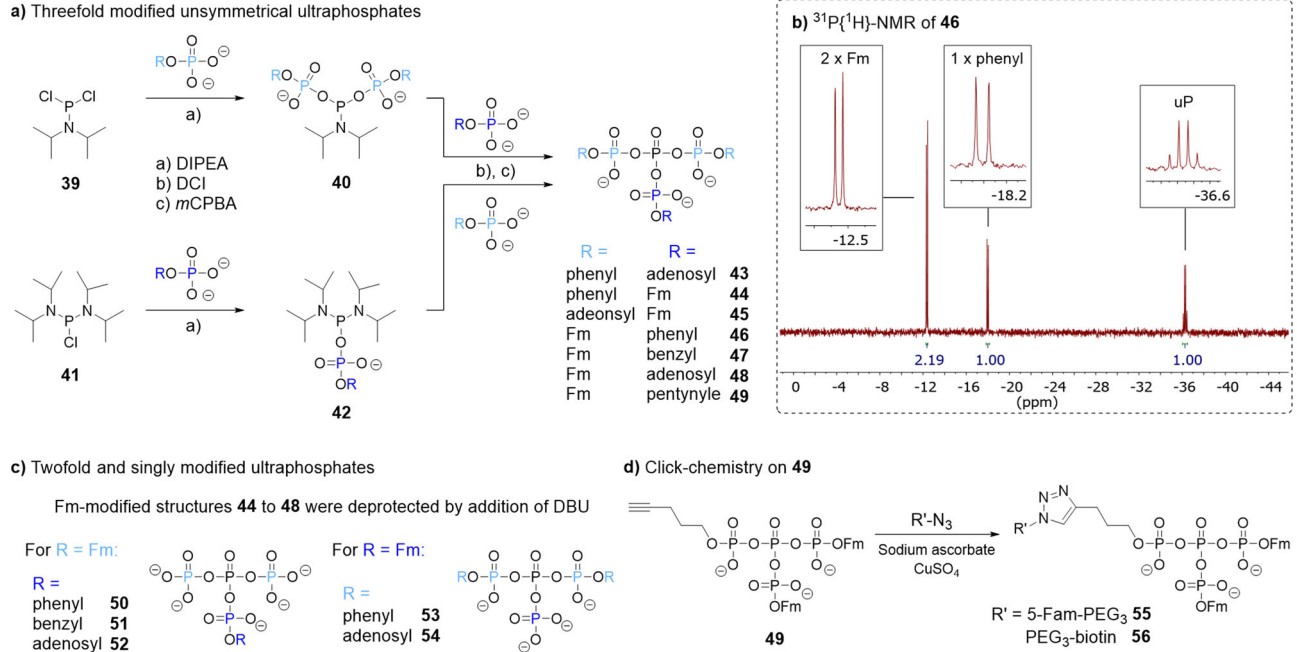

**Fig. 3 Unsymmetrical ultraphosphates.** For the discussion of yields see Supplementary Methods 12.2. **a** Syntheses of threefold modified unsymmetrical ultraphosphates using either a chloro phosphormono- or diamidite. **b** $^{31}$P{$^1$H}-NMR spectrum of unsymmetrically modified ultraphosphate **46**. **c** Twofold and singly modified ultraphosphates by deprotection of Fm-modified, unsymmetrical ultraphosphates using DBU. **d** Application of click-chemistry on ultraphosphate **49**.

modifications influenced the stability of the ultraphosphate towards hydrolysis, showing faster decomposition for twofold modified structures. For example, **54** was completely hydrolysed in the deprotection mixture (5% DBU in water, pH 13.9) within 30 min while **52** had a half-life of 180 min. Copper-catalysed click reactions on **49** were performed with biotin and fluorescein-conjugates and allowed the preservation of the ultraphosphate (Fig. 3d).

**Salt metatheses and crystallisation**. To obtain a single crystal of an ultraphosphate, we examined salt metatheses reactions to yield [PPN] salts, as [PPN] is well-known for its high crystallinity and its capability to stabilise anions[39]. We developed a method to isolate the ultraphosphate [PPN] salts from fractions after SAX purification (Supplementary Fig. 2) and found that they can be dried and stored indefinitely, in contrast to non-[PPN] salts. Only uP$_4$ [PPN] decomposed as soon as residual water was removed.

Although no crystals of sufficient quality for crystal structure analysis could be obtained from the [PPN] salts, it was found that Fm-protected **37** can be salted out from solutions after reversed-phase purification (55% water, 35% MeCN, 10% TEAA (100 mM, pH 7.0)) by addition of NaCl and that these conditions supported the formation of single-crystals. X-ray diffraction studies established the identity of **37** (Fig. 2). The packing along the a-axis (Supplementary Fig. 3) shows that the ultraphosphate is arranged along a chain of water-bridged sodium cations to form distinct layers where hydro- and lipophilic interactions are maximised. We attribute the increased stability of **37** compared to other symmetrically modified ultraphosphates to a steric shielding of the ultraphosphate moiety. Accordingly, the comparably lower stability of singly or twofold modified ultraphosphates can be explained. However, as a counter-effect, the higher negative charge enables longer half-lives of singly compared with twofold modified structures. Although [PPN] salts are stable storage forms for ultraphosphates, their insolubility in water can be disadvantageous. The metathesis to water-soluble sodium salts

was possible using NaOTf. Only for uP$_4$, the salt metathesis was unsuccessful and led to the formation of inorganic mono- and triphosphate. Unexpectedly, also pyro- (14%) and linear tetraphosphate (12%) were detected.

**Stability of ultraphosphates in water**. The antibranching-rule describes ultraphosphates as exceedingly unstable in the presence of water[12, 13]. Our observations, however, revealed significant hydrolytic stability for short-chain ultraphosphates expressed in half-lives ranging from several hours to days. The decomposition of uP$_4$ and trisadenosine uP$_4$ **21** was tracked by $^{31}$P{$^1$H}-NMR at different pH values and in the presence of different cations (see Fig. 4a and Supplementary Fig. 4). There is a high pH-dependency on the decomposition rates, with longer half-lives obtained with increasing pH values. In the presence of Mg$^{2+}$, the decay was about twice as fast, and Ca$^{2+}$ accelerated the decay even further. In contrast, monovalent cations had only little effect.

The stability of ultraphosphates suffices to subject them to polyacrylamide gel electrophoresis on dense gels (PAGE; Supplementary Fig. 5 and Supplementary Fig. 6) without significant decomposition followed by staining with toluidine blue[40]. Such separations take several hours in aqueous buffer. Threefold modified ultraphosphates gave clear bands without any observed decay and even uP$_4$ could be analysed by PAGE, pointing towards a possible analytical approach to detect ultraphosphates in biological samples. Singly or twofold modified structures **52** and **54** decomposed under these conditions.

**Enzymatic digestion of ultraphosphates**. Ultraphosphates have not been reported to occur in biological systems[41]. Since polyP extraction protocols usually include acidic conditions, nucleophilic reagents, divalent cations and drying—all of which accelerate ultraphosphate decomposition—one would not expect to detect ultraphosphates. The publication by Hong provides evidence for this case already for the more stable metaphosphates[9].

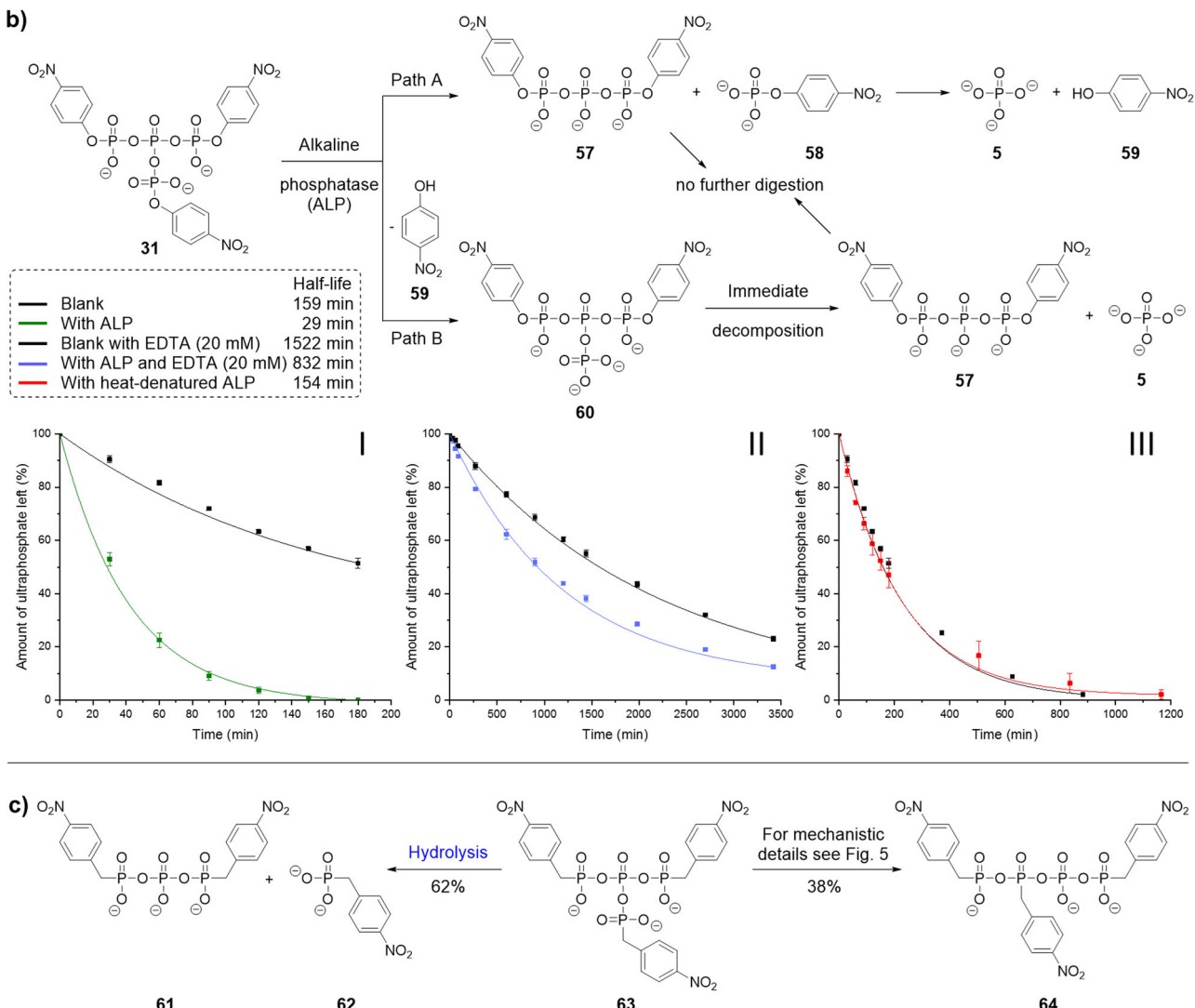

**Fig. 4 Stability of ultraphosphates in aqueous media and enzymatic digestion. a** Decomposition of trisadenosine ultraphosphate (**21**) and uP$_4$ under different pH values and in presence of 0.1 eq. of different cations. Half-lives were calculated assuming pseudo-first-order reaction kinetics. **b** Enzymatic digestion of tris(*para*-nitrophenyl) ultraphosphate (**31**) by alkaline phosphatase (ALP) from bovine intestinal mucosa. Kinetics were recorded using $^{31}$P{$^1$H}-NMR and the [PPN] signal was used as an internal standard. The results are means ± standard deviation from experiments performed in triplicates. Half-lives were calculated assuming pseudo-first-order reaction kinetics. **I**, **31** in the presence (green) and absence (black) of ALP. **II**, **31** with 20 mM EDTA in the presence (blue) and absence (black) of ALP. **III**, **31** in the presence (red) and absence (black) of heat-inactivated ALP. **c** Decomposition of tris(*para*-nitrobenzyl phosphonyl) ultraphosphate (**63**).

In accordance, we could not find any ultraphosphate signal at ca. $\delta = -35$ ppm in $^{31}$P{$^1$H}-NMR analyses of polyP extracts from different yeast strains modified in their polyP metabolising enzymes (the polyP overexpressor GPD-vtc5, the polyP devoid Δ vtc4, and polyP accumulating Δ ppn1, ppn2, ppx1, that lack phosphatases degrading polyP; Supplementary Fig. 7 and 8). We hypothesised that in addition to chemical conditions, enzymes might also degrade ultraphosphates, which would further

complicate strategies to extract them from biological sources. We studied the enzymatic digest of ultraphosphates using alkaline phosphatase (ALP). We initially examined the hydrolysis of para-nitrophenylphosphate and its ultraphosphate **31**. We synthesised ultraphosphate **31** as its [PPN] salt and tracked the decomposition in the presence and absence of ALP by $^{31}$P{$^1$H}-NMR (Fig. 4b)[42]. **31** showed a half-life of $t_{1/2} = 159$ min in the absence of the enzyme, which was significantly reduced upon its addition

($t_{1/2}$ = 29 min). We were interested as to whether the ultraphosphate **31** binds to the active site or whether surface effects are responsible for the accelerated hydrolysis. We measured additional kinetics using either heat-denatured enzyme or enzyme-treated with ethylenediaminetetraacetic acid (EDTA) to remove $Mg^{2+}$ and $Zn^{2+}$ ions. The decay of **31** in presence of heat-inactivated enzyme copies the kinetics of the blank. Only a low residual hydrolytic activity in the presence of EDTA was found. We conclude that ultraphosphate **31** binds to the active site and is enzymatically digested, providing evidence for an ultraphosphate–enzyme interaction. Similar results were obtained for the adenosine-modified ultraphosphate **21** (Supplementary Fig. 9). However, it was still unclear whether ALP first hydrolyses the branching point of ultraphosphates, or one of the three phosphoester bonds (path A or B in Fig. 4b). The latter would result in an unstable twofold modified ultraphosphate **60** that would rapidly decay further. uP$_4$ **2** was enzymatically digested as well (Supplementary Fig. 10), pointing towards the cleavage of the anhydride bond. To further study the enzymatic digestion of modified ultraphosphates, we envisioned the synthesis of the nonhydrolyzable analogue **63** with a $CH_2$-group as an oxygen replacement, which the enzyme cannot cleave, whereas still being capable of hydrolysing the central anhydrides (Fig. 4c). *para*-Nitrobenzyl phosphonic acid was transformed to ultraphosphonate **63**. After oxidation, $^{31}P\{^1H\}$-NMR suggested a clean reaction to the product, but after precipitation, ultraphosphonate **63** was no longer present (Supplementary Fig. 11). We identified the usual hydrolysis products **61** and **62** but also another product, which was purified by AIEX chromatography and found to be a linearised mixed tetraphosphonate/phosphate analogue **64**. This is reminiscent of the linear tetraphosphate we found for uP$_4$ **2** decomposition described earlier and pointed towards a rearrangement typical for ultraphosphates.

### Ultraphosphates rearrange by phosphate walk

Apart from linearised products such as P$_4$ from uP$_4$ [PPN] **2** and the synthesis of an ultraphosphonate (**63** and **64**, Fig. 4c and Supplementary Fig. 11), trisadenosine ultraphosphate [PPN] (**21**) also showed another decay mechanism accompanying simple hydrolysis of the branching point after several days in DMF. Next to AMP and Ap$_3$A as expected products of hydrolysis, further signals were detected and $^1H$-$^{31}P$-HMBC cross-peak analysis revealed that internal phosphates must carry adenosine (Fig. 5a). The mixture was analysed by CE-MS, which enabled the separation of different components and determination of their mass. We confirmed masses matching Ap$_2$A, a threefold modified triphosphate **65** as well as a threefold modified tetraphosphate **66**. Since **21** was analytically pure, a rearrangement must lead to linearization of the ultraphosphate that walks into the line[43], which we dub the phosphate walk (Fig. 5b). We propose, that the ultraphosphate linearises by an attack of one terminal phosphate at another accompanied with the cleavage of either a phosphoanhydride or phosphoester bond. The latter would form a cyclic ultraphosphate intermediate and requires a second nucleophilic attack of the released alcoholate to form **66** as the central intermediate of the mechanism. Depending on the nucleophile (AMP or $H_2O$) and the attacked phosphate of **66** (Fig. 5b), both the internally modified phosphate **65** and Ap$_2$A, as well as standard decomposition products, can be explained.

Bisphenyladenosine (**43**) and trisbenzyl ultraphosphate (**30**) [PPN] (Supplementary Fig. 12 and Supplementary Fig. 13) also showed phosphate walk products. For **30**, the rearrangement could only be induced by heating to 80 °C.

Computation was invoked to corroborate the feasibility of the internal phosphate walk mechanism. The electronic states and chemical bonding of branched phosphates in phosphate glasses have been discussed based on computational results for various model clusters[44]. However, the history of mechanistic disputes surrounding substitution reactions at phosphorous includes a myriad of experimental results and ab initio computations[45–47]. The energy of highly charged species, with variable counterions in high vs moderate dielectrics involving hydrogen bond donors, are sensitive to intricate changes in speciation, constitutional and conformation isomerism, and explicit vs continuum environmental effects[47–49]. Mechanistic computational studies involving trimethyl ultraphosphate (**67**) at the B97D/Def2-TZVPD(water) level of theory support the ultraphosphate **67** and linearised product **69** having energies compatible with both species being accessible under normal conditions. A cyclic intermediate **68** (Fig. 5c) was found that results formally from the loss of methoxide and the formation of a six-membered ring by the association of formally negative oxygen of one phosphate to the phosphorous where methoxide is removed. These three states (**67**–**69**) were the basic minima considered. The walk could occur through this higher energy cyclic state that serves as a possible bridge between branched and linear forms and has transition states leading to the ring-formation and -opening. Alternatively, the association could form a ring with a pentavalent phosphorous atom (**70**); however, numerous attempts to identify such an intermediate by DFT calculations were unsuccessful. Mono- and dihydrates were considered, where the explicit waters were placed in positions consistent with prevailing models[47] bringing down the relative activation energies. The data support a trend where reaction in water coupled to specific acid catalysis would show fast substitution rates. Considering effective molarities, relative nucleophilicities and the high concentration of water in water (55 M), hydrolysis as the dominant reaction path is not surprising. Taken out of the context of water, the effective molarity of internal nucleophile remains constant; the nucleophilic strength of solvent is reduced and the concentration of solvent molecules as nucleophiles in bulk solvent drops to on the order of 10 M[50, 51]. All these factors presage parallel rates of acceleration in the presence of acid but a product distribution in which internal attack becomes competitive if not fully dominant. This shift in product distribution opens the way for the phosphate walk. For the ultraphosphonate analogue **63**, however, the one-step mechanism must be energetically favoured.

### Reactivity of uP$_4$ as phosphorylating agent

Ultraphosphates could potentially serve as phosphorylating reagents if suitable nucleophiles are present, as has been previously studied for poly- and metaphosphates[52–55]. Earlier studies suggested the formation of branched phosphates on primitive earth as partial hydrolysis products of $P_4O_{10}$, which in turn can be volatilised from magma[7, 56–58]. uP$_4$ **2** was used as a model substrate for polydisperse ultraphosphates arising from $P_4O_{10}$ in the presence of water[59] to study the potential contribution of branched phosphates to prebiotic phosphorylation reactions (Fig. 6a and Supplementary Fig. 15). Aliphatic nucleophiles were studied for general reactivity patterns of ultraphosphates. Ethanolamine was used to screen the required stoichiometry to favour the phosphorylation reaction over simple hydrolysis by water. We observed monophosphorylation of the amine moiety with a phosphorylation ratio of up to 85:15 (phosphorylated ethanolamine vs. orthophosphate as hydrolysis product) with 3000 eq. ethanolamine. Decreasing ethanolamine gave ratios of 70:30 for 500 eq. and 27:73 phosphorylated ethanolamine for 100 eq., respectively. For secondary amines and alcohols,

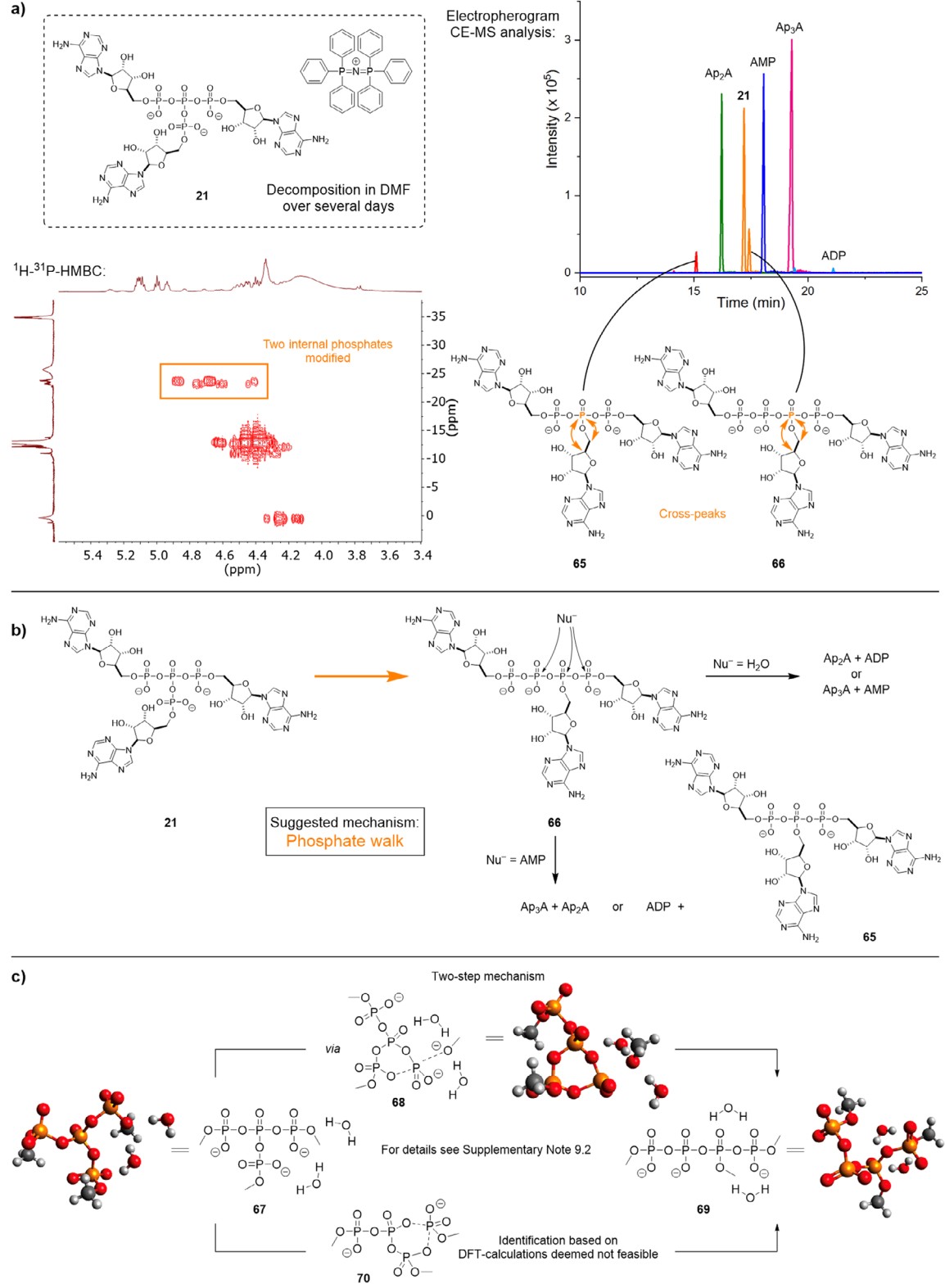

**Fig. 5 Rearrangement of ultraphosphates. a** Analytical results for trisadenosine ultraphosphate (**21**) [PPN] salt after 14 days in DMF: $^1$H-$^{31}$P-HMBC with two cross-peaks for internally modified oligophosphates. CE-MS analysis of the product mixture and proposed structures **65** and **66** for the internally modified compounds. **b** Phosphate walk rearrangement for trisadenosine ultraphosphate (**21**) and nucleophilic attack of the linearised product **66**. **c** Possible mechanistic pathways of the phosphate walk rearrangement for trimethyl ultraphosphate dihydrate (**67**). Mechanistic studies were carried out at the B97D/Def2-TZVPD(water) level of theory; for details see Supplementary Note 9.2.

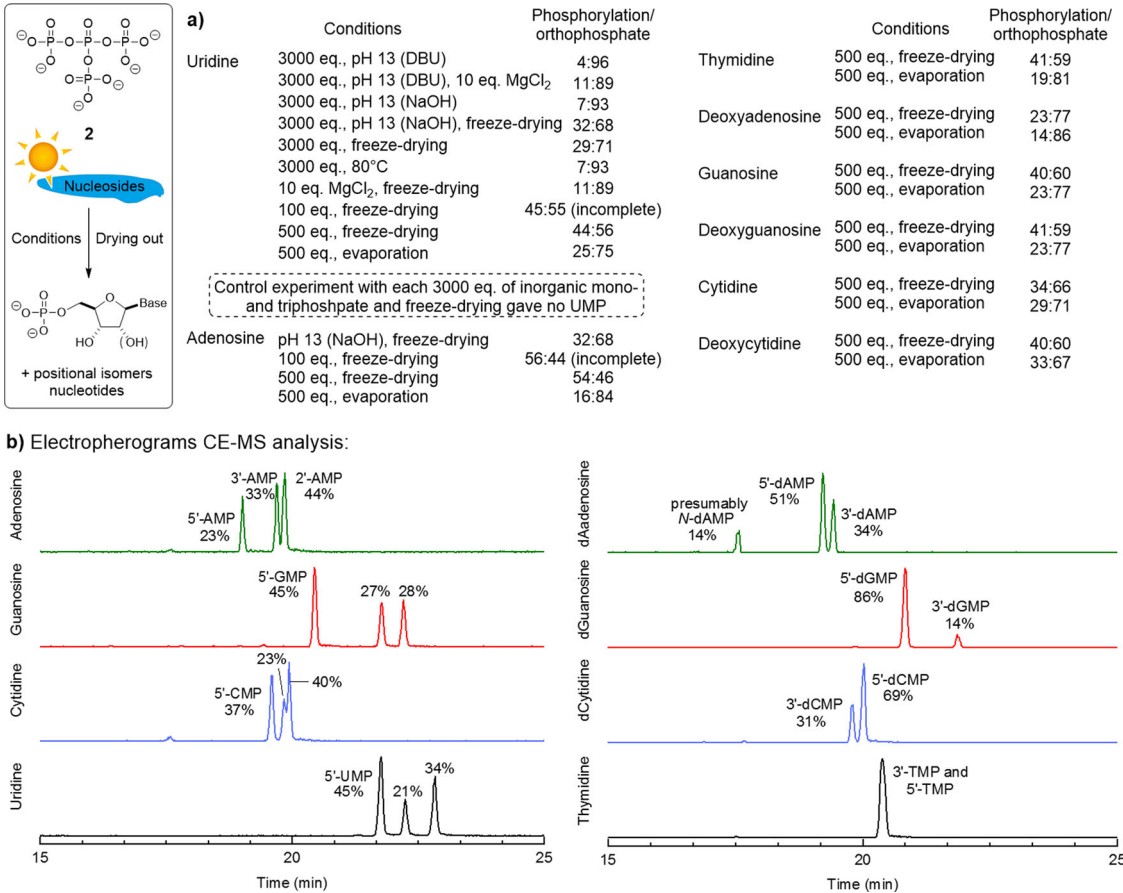

**Fig. 6 Phosphorylation of nucleosides by uP₄ (2) and electropherograms of the CE-MS analysis. a** Conditions and phosphorylation rates for the reaction of nucleosides with uP₄. If the pH of the nucleophile solution was adjusted, the applied base is indicated in brackets. Phosphorylation rates were calculated without consideration of the phosphoramidate by-product arising from the reaction of DBU with uP₄. **b** Electropherograms of the CE-MS analysis of the product distribution. The positional isomers were determined by spiking experiments.

we additionally detected a phosphoramidate arising from the reaction of DBU — which was present due to the deprotection conditions —with uP₄. This side-reaction was avoided using polymer-bound DBU for the deprotection, affording DBU free uP₄ in 81% purity (see Supplementary Fig. 15). Interestingly, uP₄ **2** was found to be less stable in the absence of DBU. Thus, further experiments were performed with DBU in solution due to the better repeatability. For different amino acids, the primary or secondary amine and — if present — the alcohol or thiol moiety were phosphorylated (Supplementary Fig. 15). For glycine, only 8% phosphorylation of the amine was observed at pH 8.8 but the adjustment to pH 13.0 increased the phosphorylation to 66%, in accordance with the longer half-life of uP₄ at higher pH (see Fig. 4a). Although it is unclear whether this pH condition is relevant to prebiotic chemistry[60], the result still shows a general reactivity trend. Freeze-drying experiments increased the phosphorylation of glycine to its phosphoramidate to 52% at a pH of ca. 9.

For all canonical (deoxy)nucleosides, even 10 eq. were sufficient to detect phosphorylated nucleosides by CE-MS after freeze-drying. For full consumption of uP₄ **2** in one single freeze-drying experiment, 500 eq. were necessary. These conditions allowed phosphorylation of up to 54%, whereas evaporation of solvent at room temperature resulted in 33% nucleoside phosphorylation. The product distribution was analysed by CE-MS, and spiking experiments revealed product identities (Fig. 6b). We found phosphorylation in the 5′- and also 2′- and 3′-positions. To rule out phosphorylation arising from inorganic mono- and

triphosphate, these compounds were lyophilised with 3000 eq. uridine, leading to no phosphorylation. Ultraphosphate **2** as a model substrate—and by extension other branched polyP—may thus have served as phosphate donors in prebiotic chemistry.

Life relies on condensed phosphates, but ultraphosphates as a fundamental type and true constitutional isomers of the linear polyP remained unstudied. In this study, we disclose the synthesis of defined monodisperse ultraphosphates containing zero to three modifications. For those short-chain ultraphosphates, we found significant hydrolytic stability expressed in half-lives up to days, which calls the antibranching-rule into question. We provide evidence for the interaction of an enzyme with ultraphosphates and describe the phosphate walk, which linearises branched phosphates. Ultraphosphate was applied as a phosphorylating reagent for nucleophiles such as amino acids and nucleosides with implications for prebiotic chemistry. With synthetic access to this class of molecules, the chemistry—and potential biology—of the branched phosphates can finally be studied.

## Methods

**General procedure for the synthesis of ultraphosphates.** In all, 3.0 eq. of a monophosphate TBA salt and 3.0 eq. of an activator (ETT or DCI) were co-evaporated using MeCN (3×) and afterwards dissolved in DMF. 1.0 eq. of P(NEt₂)₃ was added and stirred for 10 min. Upon completion of the coupling, 1.5 eq. mCPBA (77%) was added at 0 °C and stirred for 10 min. The product was pre-cipitated using Et₂O, the suspension centrifuged and the pellet washed with Et₂O. Drying in vacuo gave the crude product as a mixed TBA/diethylammonium salt. The ultraphosphate was purified by AIEX chromatography (Q Sepharose® Fast

Flow) or using a PuriFlash Column (30 µ C18 AQ; water, MeCN gradient (0–45%), 10% TEAA (100 mM, pH 7.0)).

### General procedure for the salt exchange of ultraphosphates to [PPN] salts.

The crude ultraphosphate is either purified by anion exchange chromatography using Q Sepharose® Fast Flow and a NaCl (1 M) gradient or by using a PuriFlash Column (30 µ C18 AQ; water, MeCN gradient (0–45%), 10% TEAA (100 mM, pH 7.0)). Fractions containing ultraphosphate were analysed for their concentration by addition of a defined volume of a PMe$_4$Br solution in D$_2$O (1 mg/ml) and $^{31}$P{$^1$H}-NMR. Fractions were combined and heated to 55 °C. [PPN]-Cl (2–3 eq.) was dissolved in H$_2$O (yielding ~2–5 mM solution) at 55 °C and added to the ultraphosphate solution. The precipitate was collected by centrifugation and washed with warm water (~55 °C). The residue was dissolved in acetone, dried over Na$_2$SO$_4$ and the solvent removed in vacuo.

### Data availability

All data supporting the findings of this study are available in the article and Supplementary Information file or from the corresponding author upon reasonable request. The X-ray crystallography data have been deposited at the Cambridge Crystallographic Data Centre (CCDC), under accession number CCDC: 2032762. These data can be obtained free of charge from The Cambridge Crystallographic Data Centre via www.ccdc.cam.ac.uk/data_request/cif.

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

## Acknowledgements

We thank Dr. Manfred Keller from MagRes of the University of Freiburg for the kind support and a significant amount of time for NMR spectroscopy. We thank Christoph Warth for HRMS measurements, Wiebke Unkrig and Stefan Meier for Raman measurements and Daniel Kratzert and Boumahdi Benkmil for support in X-ray crystallography. We acknowledge helpful discussions with Scott M. Shepard and Christopher C. Cummins regarding crystallisation conditions. Moreover, we thank Thomas Haas, Jyoti Singh and Laura Mayer for their insightful comments. The authors gratefully acknowledge financial support from the VW Foundation (Experiment!AZ92270), and the Human Frontier Science Program HFSP (RGP0025/2016). This study was supported by the Deutsche Forschungsgemeinschaft (DFG, German Research Foundation) under Germany's Excellence Strategy (EXC-2193/1-390951807) and a scholarship grant from Cusanuswerk.

## Author contributions

T.D-M. and H.J.J. conceived the research project and wrote the manuscript. T.D. prepared the samples and applied them in follow-up studies. N.S. and A.H. gave input on the synthetic procedure. D.Q. developed the CE-MS method. D.Q. and M.H. analysed samples using CE-MS and supported data analysis. V.B.E. analysed samples using PAGE. A.M. provided polyP extracts from yeast. J.S.S. and K.K.B. performed DFT calculations.

## Funding

## Competing interests

The authors declare no competing interests.
