## [Peer Review File · Nature Communications]

Reviewers' Comments:

Reviewer #1:

Remarks to the Author:

The authors revisit the synthesis and as they were successful, the characterization of branched polyphosphate. The existence of branched polyP in nature is not confirmed, but for sure the synthesized material was significantly more stable in water than expected from the literature. The manuscript is carefully written. I would like to ask the authors to recheck all graphs, as I found some challenges.

Manuscript:

Figure 2a (i): I would guess that these are the signals of the reactants 16 and 17 and not the intermediate. This would also explain the missing signal splitting.

Supplementary part:

The spectrum on page 76, for example, is somewhat misleading as it shows significantly more signals than would be expected for component 25. Component 25 is included at most as a by-product. Have the other signals been assigned?

Reviewer #2:

Remarks to the Author:

In this work, Jessen and coworkers have performed a detailed analysis of the chemistry of branched condensed phosphates, which is a biochemical / bioinorganic chemistry problem that has received far too little attention by the field, and therefore the manuscript is timely and interesting. I enjoyed reading the manuscript, and thought the phosphate walk mechanism was compelling; however, there are a number of issues (some minor, some less so) that need to be addressed with regard to the computational component of the work. Specifically:

1. As a visual issue, for Supplementary Fig. 14, could you please use a white background for the structures, and ideally label also key distances on the structures to track with the energies? Also, two decimal places is excessive for the energies.
2. The computational methods in the SI are very brief, more detail would be helpful. For example, are these continuous stationary points, obtained by following IRC from transition states, or where they optimized separately? How do the authors ensure the profiles are continuous and the stationary points linked? Also, what software was used for the simulations?
3. Can the authors provide coordinates of all stationary points computationally characterized in a format that can be read by commonly used quantum chemistry packages, as well as a table of the absolute energies and vibrational frequencies? (also can the authors add free energy corrections to show dG rather than dH please?)
4. It's unclear (see e.g. prior work on phosphate monoesters/polyphosphate hydrolysis) whether a pure continuum model accurately describes the geometries and energies of the key species involved in phosphoryl transfer reactions, and explicit hydrogen bonding interactions to water molecules in a mixed solvent model can make a huge difference. While I agree with the authors that it's very plausible that indeed the one-step mechanism shown in Figure 5 does not occur, it would be helpful to validate that this is not just an artefact by including also explicit water molecules in the optimization.
5. The dH values shown in Supplementary Fig. S14 are extremely high, and not consistent with the claim of the authors on pg. 15 of the main text that both species 67 and 69 are accessible under normal conditions, as with dH values that high the interconversion will be extremely slow. This in turn contradicts the

experimental observation on pg. 13 that the decay mechanism accompanying simple hydrolysis of the branching mode was observed after several days in DMF. Likely there is a problem with the calculations, but based on the amount of information provided (see points 2/3) it's hard to know what. Can the authors please investigate and address these discrepancies?

Reviewer #3:

Remarks to the Author:

This is a very thorough and well documented paper on the synthesis and characterization of branched oligophosphates, or 'ultraphosphates'. A synthetic methodology for the preparation of novel defined branched phosphate species is described, along with detailed characterization, information on various protecting groups and deprotection methods, and data on the stability of the branched products to hydrolysis. And interesting rearrangement to linear oligophosphates is described, along with experiments on the use of these reactive phosphates for prebiotically relevant phosphorylation reactions. The article is suitable for publication with only the following extremely minor corrections:

Fig. 2, the structure of compound 19 should be defined in the figure.

p. 9, line 141, spelling: definitely

p. 12, line 209 (also 234), should phosphorester be phosphoester?

p. 15, line 256, viz should be vs.

This reviewer appreciated ref. 48.

Reviewer: Jack W. Szostak

Reviewer #4:

Remarks to the Author:

The major claim of this paper is the first successful synthesis of monodisperse short-chain, branched polyphosphates 'ultraphosphates'. In the 50s, this class of compounds was identified as less stable than linear polyphosphates (or cyclic polyphosphates; 'metaphosphates' strictly applies to cyclic derivatives of metaphosphoric acid (PO₃H) which exists stably as a trimer). Thus, an 'anti-branching rule' was posited on the elusiveness of ultraphosphates. The authors have applied phosphorotriamidite chemistry with monophosphate esters to obtain ultraphosphite triesters, which they then conventionally oxidize with mCPBA to prepare corresponding ultraphosphates. I found this study to be a rather systematic tour de force of heroic synthetic and analytical chemistry that makes available for the first time, at least crude samples of a long-sought class of ephemeral phosphate derivatives. Besides asserting a refutation of the 'anti-branching rule', the authors argue for a possible significance to prebiotic phosphorus chemistry and offer evidence for catalyzed decomposition with a phosphatase, increasing the potential interested readership range beyond chemists.

The paper is well-organized and written clearly. Given the lability of the target compounds, the structural characterizations are largely based on NMR, MS and in one case a crystal structure of a PPN salt.

The authors should consider responding to the following points:

1. Have they really refuted, or in a sense, affirmed the 'anti-branching rule'? The parent compound, uP4 was obtained in solution but could not be isolated; its PPN salt "decomposed as soon as residual water was removed". Like the parent, the lower esters proved unstable in aqueous media and in general, only relatively impure products could be obtained due to lability.
2. They present DFT calculations, but do not mention/compare the much earlier calculations of Kowada et al.

on ultraphosphate glasses claiming to rationalize the antibranching rule for these cognate species (or more recent embodiments). *J Phys Chem* 97, 8989 (1993)

3. The enzyme studies would be strengthened by determining kinetic parameters (K_m , k_{cat}) in the presence and absence of a known phosphatase inhibitor with determination of the type of inhibition by the latter.

4. With respect to phosphorylation properties of synthetic uP4/derivatives and their relevance to prebiotic nucleotide synthesis, it is interesting that a 5'-Ado ester of uP4 could be made, but the significance of phosphoramidite chemistry under prebiotic conditions is not clear. Due to the great lability of uP4s, they will not be found in nature archeologically. By contrast, this is not true of cyclophosphates which have been identified as natural minerals. Britvin et al., *Geology* <https://doi.org/10.1130/G48203> (2020). Discuss relevance more convincingly.

5. The phosphate chemistry review articles cited are not very modern. I commend to the authors' attention a 2021 review, 42.16.4 Phosphoric Acid and its Derivatives by Kashemirov et al. Chapter_42.16.4 in *Science of Synthesis* (Thieme Verlag) which can be cited as confirming the originality of their work (no synthesis of uL4s is documented from the reviewed literature).

Prof. Dr. Henning J. Jessen

Director
Institute of Organic Chemistry
Bioorganic Chemistry
Albert-Ludwigs-University
Albertstrasse 21, D-79104 Freiburg

Tel. 0761/203-6073

Tel. 0761/203-6041
Secretary: Regine Schandera

RE: Nature Communications manuscript NCOMMS-21-13700-T

henning.jessen@oc.uni-freiburg.de
<http://www.jessen-lab.uni-freiburg.de>

2nd of July 2021

Dear Reviewers

thank you for your overall positive comments on our manuscript "**The Chemistry of Branched Condensed Phosphates**"

We appreciate the input and provide detailed answers and comments below (in italic).

REVIEWER COMMENTS

Reviewer #1 (Remarks to the Author):

The authors revisit the synthesis and as they were successful, the characterization of branched polyphosphate. The existence of branched polyP in nature is not confirmed, but for sure the synthesized material was significantly more stable in water than expected from the literature. The manuscript is carefully written. I would like to ask the authors to recheck all graphs, as I found some challenges.

- *We thank the reviewer for these positive comments. We have checked the graphs and made minor corrections.*

Manuscript:

Figure 2a (i): I would guess that these are the signals of the reactants 16 and 17 and not the intermediate. This would also explain the missing signal splitting.

- *The chemical shift values indicate that these signals represent the P(III)-P(V) anhydride, as the reactants are consumed. The absence of peak splitting has been observed by us before. For example, the terminal phosphate (blue) in intermediate 18 resonates at -10 ppm, typical for an anhydride. The very small signal remaining at ca. 0 ppm is from the reactant 17. We are therefore confident that our assignment is correct.*

Supplementary part:

The spectrum on page 76, for example, is somewhat misleading as it shows significantly more signals than would be expected for component 25. Component 25 is included at most as a by-product. Have the other signals been assigned?

- *We apologize for the omission. We wanted to focus on the ultraphosphate structure and have not indicated the counterion in all spectra. The signals in the aromatic region are caused by the PPN counterion. We have now indicated the PPN counterion in the spectra.*

Reviewer #2 (Remarks to the Author):

In this work, Jessen and coworkers have performed a detailed analysis of the chemistry of branched condensed phosphates, which is a biochemical / bioinorganic chemistry problem that has received far too little attention by the field, and therefore the manuscript is timely and interesting. I enjoyed reading the manuscript, and thought the phosphate walk mechanism was compelling; however, there are a number of issues (some minor, some less so) that need to be addressed with regard to the computational component of the work.

- *We thank the reviewer for this very positive assessment of our work.*

Specifically:

1. As a visual issue, for Supplementary Fig. 14, could you please use a white background for the structures, and ideally label also key distances on the structures to track with the energies? Also, two decimal places is excessive for the energies.

- *This has been done and coordinates are now provided for these structures so that they can be analyzed in detail for all internal geometries.*

2. The computational methods in the SI are very brief, more detail would be helpful. For example, are these continuous stationary points, obtained by following IRC from transition states, or where they optimized separately? How do the authors ensure the profiles are

continuous and the stationary points linked? Also, what software was used for the simulations?

- *These are optimized minima (zero negative eigenvalues) and TS's (one negative eigenvalue). IRCs confirm the connection between TS and minima. Details have been added to the computational methods in the SI.*

3. Can the authors provide coordinates of all stationary points computationally characterized in a format that can be read by commonly used quantum chemistry packages, as well as a table of the absolute energies and vibrational frequencies? (also can the authors add free energy corrections to show dG rather than dH please?)

- *These details have been added to the computational methods.*

4. It's unclear (see e.g. prior work on phosphate monoesters/polyphosphate hydrolysis) whether a pure continuum model accurately describes the geometries and energies of the key species involved in phosphoryl transfer reactions, and explicit hydrogen bonding interactions to water molecules in a mixed solvent model can make a huge difference. While I agree with the authors that it's very plausible that indeed the one-step mechanism shown in Figure 5 does not occur, it would be helpful to validate that this is not just an artefact by including also explicit water molecules in the optimization.

- *The one-step mechanism has been analyzed in detail using zero, one and two explicit solvent molecules, however, this process still does not appear to be a viable alternative. Placing explicit waters in a system with so many oxygen atoms is too computationally intensive to be done exhaustively in a tractable time. Implementation of our sector model previously used for carboxylate acidity focused our efforts and allowed us to get reasonable structures efficiently.*

5. The dH values shown in Supplementary Fig. S14 are extremely high, and not consistent with the claim of the authors on pg. 15 of the main text that both species 67 and 69 are accessible under normal conditions, as with dH values that high the interconversion will be extremely slow. This in turn contradicts the experimental observation on pg. 13 that the decay mechanism accompanying simple hydrolysis of the branching mode was observed after several days in DMF. Likely there is a problem with the calculations, but based on the amount of information provided (see points 2/3) it's hard to know what. Can the authors please investigate and address these discrepancies?

- *The review's comments are rigorously correct for systems in which one has an expectation of high precision in the computational values; however, as we mention in the discussion, small changes in energy can lead to large changes in rate, and several factors*

could change the absolute energy scale of the potential energy surface. Ionic strength, explicit solvation, specific coordination and other factors could influence the absolute rate. The computational model presented here is a heuristic of the reaction progress in general. The trend from zero, one, and two explicit waters indicates that the relative TS energy comes down while the qualitative features do not change. The extrapolation is made that for additional waters and consideration of ionic strength, the computational numbers would converge well with the experimental results. We have changed our introductory statement in this section to "Mechanistic computational studies involving trimethyl ultraphosphate (67) at the B97D/Def2-TZVPD(water) level of theory support the ultraphosphate 67 and linearized product 69 having energies compatible with both species being potentially accessible under normal conditions in the presence of increasing number of water molecules." We think that this better reflects the intention and outcome of the calculations and thank the reviewer for helping us to clarify this part.

Reviewer #3 (Remarks to the Author):

This is a very thorough and well documented paper on the synthesis and characterization of branched oligophosphates, or 'ultraphosphates'. A synthetic methodology for the preparation of novel defined branched phosphate species is described, along with detailed characterization, information on various protecting groups and deprotection methods, and data on the stability of the branched products to hydrolysis. And interesting rearrangement to linear oligophosphates is described, along with experiments on the use of these reactive phosphates for prebiotically relevant phosphorylation reactions. The article is suitable for publication with only the following extremely minor corrections:

- *We thank the reviewer for this very positive assessment of our work.*

Fig. 2, the structure of compound 19 should be defined in the figure.

- *We added "Trisadenosin ultraphosphite" to figure 2.*

p. 9, line 141, spelling: definitely

- *Changed accordingly*

p. 12, line 209 (also 234), should phosphorester be phosphoester?

- *Changed accordingly*

p. 15, line 256, viz should be vs.

- *Changed accordingly*

This reviewer appreciated ref. 48.

Reviewer: Jack W. Szostak

Reviewer #4 (Remarks to the Author):

The major claim of this paper is the first successful synthesis of monodisperse short-chain, branched polyphosphates 'ultraphosphates'. In the 50s, this class of compounds was identified as less stable than linear polyphosphates (or cyclic polyphosphates; 'metaphosphates' strictly applies to cyclic derivatives of metaphosphoric acid (PO_3H) which exists stably as a trimer). Thus, an 'anti-branching rule' was posited on the elusiveness of ultraphosphates. The authors have applied phosphorotriamidite chemistry with monophosphate esters to obtain ultraphosphite triesters, which they then conventionally oxidize with mCPBA to prepare corresponding ultraphosphates. I found this study to be a rather systematic tour de force of heroic synthetic and analytical chemistry that makes available for the first time, at least crude samples of a long-sought class of ephemeral phosphate derivatives. Besides asserting a refutation of the 'anti-branching rule', the authors argue for a possible significance to prebiotic phosphorus chemistry and offer evidence for catalyzed decomposition with a phosphatase, increasing the potential interested readership range beyond chemists.

The paper is well-organized and written clearly. Given the lability of the target compounds, the structural characterizations are largely based on NMR, MS and in one case a crystal structure of a PPN salt.

- *We thank the reviewer for this very positive assessment of our work.*

The authors should consider responding to the following points:
1. Have they really refuted, or in a sense, affirmed the 'anti-branching rule'? The parent compound, uP_4 was obtained in solution but could not be isolated; its PPN salt "decomposed as soon as residual water was removed". Like the parent, the lower esters proved unstable in aqueous media and in general, only relatively impure products could be obtained due to lability.

- *The reviewer clearly raises an interesting point. The initially coined antibranching rule was maybe not meant as strict. Over the decades, it has turned into something that precludes the existence of branched phosphates in water (due to "instantaneous" hydrolysis). However, the name "antibranching rule" in itself appears to be very apodictic and is, as we are convinced, misleading. One can argue that no one has ever put a "number" on the hydrolysis rates; what is meant by "extremely rapid" hydrolysis? There will be differences in answers to that question by different scientists. A geochemist or a femtosecond spectroscopist have different views on what is "extremely rapid". However, during our studies it has become quite clear that branched phosphates can be handled and*

purified in water. Actually, only the removal of residual water led to degradation. In water, some of these compounds were stable for days.

2. They present DFT calculations, but do not mention/compare the much earlier calculations of Kowada et al. on ultraphosphate glasses claiming to rationalize the antibranching rule for these cognate species (or more recent embodiments). J Phys Chem 97, 8989 (1993)

- *We have added in the computational section: The electronic states and chemical bonding of branched phosphates in phosphate glasses have been discussed based on computational results for various model clusters.⁵¹*

3. The enzyme studies would be strengthened by determining kinetic parameters (K_m , k_{cat}) in the presence and absence of a known phosphatase inhibitor with determination of the type of inhibition by the latter.

- *We thank the reviewer for this suggestion. This experiment was included to demonstrate effects of phosphatases on ultraphosphates, as one may want to isolate these structures from biological sources. This experiment indicates that phosphatase inhibition will be required in such efforts. The determination of K_m , k_{cat} is difficult for these substrates, as there is a significant background hydrolysis caused by the presence of divalent metal cations. Furthermore, the products of the first hydrolysis will be further hydrolysed by the enzyme leading to very complex behavior. A precise determination of the requested values would require an amount of work that we believe is well beyond the scope of this manuscript.*

4. With respect to phosphorylation properties of synthetic uP4/derivatives and their relevance to prebiotic nucleotide synthesis, it is interesting that a 5'-Ado ester of uP4 could be made, but the significance of phosphoramidite chemistry under prebiotic conditions is not clear.

- *We agree with the reviewer that phosphoramidite chemistry is likely not relevant under prebiotic conditions. These reagents were used to access a previously inaccessible group of molecules, but they do not contain the P-amidite anymore. Potential formation of ultraphosphates was discussed in figure 1 by partial hydrolysis of P4O10 from volcanic activity.*

Due to the great lability of uP4s, they will not be found in nature archeologically. By contrast, this is not true of cyclophosphates which have been identified as natural minerals. Britvin et al., Geology <https://doi.org/10.1130/G48203>(2020). Discuss relevance more convincingly.

- *As stated in our discussion, P4O10 could be a precursor to ultraphosphates; we agree that too little is known about this substance class and future studies should now be addressing those questions. Spontaneous production of P4O10 by volcanic activity seems plausible, as also polyphosphates are accessible under such conditions in the presence of water (Nature **352**, 516–519 (1991)). Finding the conditions that are required to form ultraphosphates directly from P4O10 or indirectly by rearrangements as described for the first time in this paper, but in the reverse direction, will be future endeavors of our group. Interestingly, the group of Cummins has very recently described the formation of a branched structure from P4O10 with DABCO in a preprint (Chemrxiv: 10.33774/chemrxiv-2021-8v8nx). Therefore, the existence of natural minerals is no strict requirement as spontaneous formation appears feasible.*

5. The phosphate chemistry review articles cited are not very modern. I commend to the authors' attention a 2021 review, 42.16.4 Phosphoric Acid and its Derivatives by Kashemirov et al. Chapter_42.16.4 in Science of Synthesis (Thieme Verlag) which can be cited as confirming the originality of their work (no synthesis of uL4s is documented from the reviewed literature).

- *We thank the reviewer for the suggestion and have added the reference (ref. 2)*
- *We have additionally added a very recent paper by Glonek (Glonek, T. Did Cyclic Metaphosphates Have a Role in the Origin of Life? Origins Life Evol. B. 51, 1–60 (2021))*

On behalf of all authors, I would like to thank you for your time and valuable input to improve this submission.

With kind regards,

Henning Jessen

REVIEWERS' COMMENTS

Reviewer #1 (Remarks to the Author):

The authors took great care and addressed all comments.

Reviewer #2 (Remarks to the Author):

The authors have addressed my concerns appropriately, I am now satisfied with the computational aspects of the manuscript.

Reviewer #4 (Remarks to the Author):

I believe the authors have responded appropriately to my critique. They should fix a typo in ref. 2 (missing a space by the date).